# Illustration of the Importance of Adjustment for within- and between-Person Variability in Dietary Intake Surveys for Assessment of Population Risk of Micronutrient Deficiency/Excess Using an Example Data Set

**DOI:** 10.3390/nu14020285

**Published:** 2022-01-11

**Authors:** Johanna H. Nel, Nelia P. Steyn, Marjanne Senekal

**Affiliations:** 1Department of Logistics, University of Stellenbosch, Stellenbosch 7602, South Africa; jhnel@sun.ac.za; 2Department of Human Biology, Health Sciences Faculty, University of Cape Town, Cape Town 7701, South Africa; nelia.steyn@uct.ac.za

**Keywords:** dietary intake, 24-h recall, micronutrients, adjustments, random variances, deficiencies

## Abstract

Nutrition intervention decisions should be evidence based. Single 24-h recalls are often used for measuring dietary intake in large dietary studies. However, this method does not consider the day-to-day variation in populations’ diets. We illustrate the importance of adjustment of single 24-h recall data to remove within-person variation using the National Cancer Institute method to calculate usual intake when estimating risk of deficiency/excess. We used an example data set comprising a single 24-h recall in a total sample of 1326 1–<10-year-old children, and two additional recalls in a sub-sample of 11%, for these purposes. Results show that risk of deficiency was materially overestimated by the single unadjusted 24-h recall for vitamins B12, A, D, C and E, while risk of excess was overestimated for vitamin A and zinc, when compared to risks derived from usual intake. Food sources rich in particular micronutrients seemed to result in overestimation of deficiency risk when intra-individual variance is not removed. Our example illustrates that the application of the NCI method in dietary surveys would contribute to the formulation of more appropriate conclusions on risk of deficiency/excess in populations to advise public health nutrition initiatives when compared to those derived from a single unadjusted 24-h recall.

## 1. Introduction

Micronutrient deficiencies have been, and still are key public health nutrition challenges faced by especially low-to-middle income countries (LMIC) [1,2,3]. Various interventions have been rolled out to address such deficiencies, of which food fortification and supplementation have been found to be cost-effective [4]. Importantly, Baye [3] cautions that inadequate and excessive micronutrient intakes may co-exist in the same country and that micronutrient interventions should be designed and targeted in such a manner that excessive intakes are avoided.

Intervention decisions should be based on documentation of evidence on the actual presence of deficiencies in a particular population, whether it is of public health concern [3], and the potential effectiveness of the intervention of choice, e.g., fortification. This evidence should ideally include nationally representative dietary survey data on usual intake to identify risk of deficiency, to quantify the nutrient gap to be filled by fortification/supplementation and to characterize risk of excessive intakes [3,5]. Biochemical data should also be collected/available to confirm the presence of nutrient deficiencies and/or excessive intakes. Collection of follow-up dietary survey data to quantify change in usual intakes and monitor risk of excess should form part of assessment of micronutrient intervention outcomes. 

Usual dietary intake is defined as the long-term average daily intake of energy and nutrients and/or foods by the National Cancer Institute (NCI): Division of Cancer Control & Population Sciences [6]. The closer dietary intake estimations are to the truth (true usual intake), the more robust the identification of risk of micronutrient deficiency or excess in populations or groups within a population [7]. Diet variation is a result of the variation within and between persons. Between-person variation is a function of the heterogeneity of a population and usual dietary patterns [8]. Within-person variation occurs due to daily fluctuations, changes in dietary intakes over weekends and special occasions, as well as seasonality. The magnitude of the mostly random within-person variance varies by nutrient and is largely dependent on cultural and ecological factors [8], as well as age, with this type of variability tending to be less in children because of lower dietary diversity [9]. Methodological challenges inherent to estimation of dietary intake such as the use of standardized recipe files and food composition databases may also contribute to within-person error [10]. These errors result in large standard errors in population groups and insignificant regression coefficients [8]. Another important result of exaggerated variation is that the percentage of subjects below or above specified cut-points will be distorted [11]. 

Basiotis et al. [12] investigated the number of days of food intake data that may be needed to estimate usual intake for selected nutrients. They concluded that if a study for example examined 100 females, a food intake record of one day would suffice to estimate true energy and iron intakes, while seven days of food intake records would be needed for estimation of vitamin A intake. If intake estimation was based on three days of food intake records per person, the sample could be reduced to 15 for energy intake, 32 for iron intake, but would need to be increased to 231 for vitamin A intake. Therefore, equally precise intakes for groups may be obtained by increasing the number of food intake records per individual, or the number of individuals in the group [12]. The ratio between the within-person and between-person variance provides further insights into the number of days needed for each nutrient; the lower the ratio, the fewer repeated measurements would be needed [8]. 

There is no gold-standard method for usual dietary intake assessment [13]. Weighed or estimated dietary records may offer a high degree of accuracy in assessing food and nutrient intake in certain settings, but can be expensive, intrusive, time-consuming and might distort dietary behavior of the respondents, especially if an enumerator is present throughout the day [7,14].

Because of these constraints, methods such as 24-h recalls and food frequencies are typically used to investigate the dietary intake of large groups [13,14,15]. Of these, a single 24-h recall has been the most frequently used because of its ease of administration and the short time over which the client has to recall food intake [14,15,16]. However, due to day-tot-day variation in the dietary intake of individuals, a single 24-h recall is a poor estimator of long-term usual intake [10,14,15,16] as it only provides an estimate of the between-person variance [17]. Collection of repeated 24-h dietary recalls, however, makes it possible to separate the within-person variance from the variance between persons and subsequent removal of intra-individual variance [17], making it possible to rank individuals according to cut-offs for inadequate or excessive intakes [14,16]. 

Repeated collection of 24-h recalls in a large sample may not always be possible in nutritional studies due to time, staff, equipment and financial constraints [17] or problems with access in informal settlements and crime ridden areas in LMIC [18]. Hence a trade-off between achieving accurate measures of usual intake and the burden for both the participant and investigator needs to be considered, especially when the research involves infants and children where parents need to be involved [9]. Dodd et al. [10] and Tooze et al. [11] suggested that dietary surveys for estimation of usual intake require two or more days of intake data from at least a subset of the population for adjustment of the single 24-h recall of the total sample using statistical modelling to distinguish and remove the within-person variation from the total variation. No clear recommendation in this regard could be obtained from the literature, besides the comment by Barbosa et al. [19] and Herrick et al. [16] that the subsample should be representative of the total population of interest. However, they did not provide any criteria for determination of representativeness. Piernas et al. [9] used a second dietary recall obtained from a subsample of 9% of children older than 6 months for modelling purposes. They found that variability was reduced for nutrient intake estimates as shown by the narrower tails of the distributions across all age groups.

A number of similar statistical modelling methods for removal of intra-individual variability using repeated 24-h have been proposed. These include the Iowa State University (ISU) method [20], the National Cancer Institute (NCI) method, Multiple Source Method (MSM) [21] and the Statistical Program to Assess Dietary Exposure (SPADE) method [22]. Simulation studies conducted by Souverein et al. [23] and Laureano et al. [13] and a comparative study by Pereira et al. [24] showed that with a few exceptions, the methods perform similarly and that differences that were present became smaller with larger sample sizes (*n* > 300). Results need to be considered with caution when data remains highly skewed after modeling and high within-person variation is present, as reflected in a high within-person to between-person variance ratio [8,13]. Laureano et al. [13] agree with Souverein et al. [23] that the choice of modeling method can be based on practical reasons such as user-friendliness or assessment of the results for making plots, simulations, or a bootstrap, as long as the sample is >300. If the sample is smaller, the MSM [21] or SPADE [22] methods should be applied. Piernas et al. [9] indicated that they used the NCI method [6] in their study to model the 24-h intake data of children, as it allows for adjustment for covariates and estimation of usual intake distributions even if repeated dietary data are available only for a sub-sample of the total. A further advantage of the NCI method is its ability to account for extreme intakes, including zero intake. 

The aim of this paper was to demonstrate the importance of adjusting 24-h recall data for within- and between-person variability while also controlling for confounders, when generating estimates of risk of micronutrient deficiency/excess for advising food fortification and supplementation intervention decisions. The data set of the 2018 Provincial Dietary Intake Study (PDIS) in South Africa [2,25] was used for these purposes. A secondary aim was to identify foods that contribute to large variations (outliers and distributions with long tails) in unadjusted (Day-1) intakes.

## 2. Materials and Methods

### 2.1. The Data Set

The target population of the data set was 1–<10-year-old children in Gauteng and the Western Cape provinces in South Africa [2,25]. These are regarded as being the two most economically active and rapidly urbanizing provinces in South Africa [26]. 

The sampling strategy of the data set incorporated a multistage stratified cluster random sampling design, using the methodology applied in demographic and health surveys as described in the Sampling and Household Listing Manual, USAID [27]. Six strata were identified during the design phase, namely two provinces (Gauteng and Western Cape), with each having three areas of residence: urban formal, urban informal and rural areas. A stratified two-stage sample design was used with a probability proportional to size sampling of enumerator areas (EAs) at the first stage, and systematic sampling of households within the EAs at the second stage. A total of 84 EAs were selected from the six strata, 25 formal residential, 10 informal residential and 11 rural EAs in Gauteng, and 18 formal residential, 10 informal residential and 10 rural EAs in the Western Cape, resulting in a total sample of 1326 children, 733 in Gauteng and 593 in the Western Cape [2]. 

### 2.2. Dietary Intake Methodology

A single 24-h recall was completed for each participant in the total sample, referred to as Day-1 intake in the remaining sections. For 1–6-year-old children, the mother/caregiver reported on the intake of the child on the previous day with no input from the child. For 7–9-year-old children, the mother/caregiver and child were interviewed together to record the dietary intake during the prior 24 h. If the child had been at a day care center the previous day, the center was visited by the fieldworker and the meals and portion sizes were determined for the 24 h in question. All weekdays and Sundays were covered proportionally. The multiple pass method of the 24-h recall was used to administer the 24-h recall [28,29].

Two additional 24-h dietary recalls were completed on a subsample of 148 (2nd recall or Day-2 intake; 11.1%) and 146 (3rd recall or Day-3 intake; 11.0%) children in the sample. For logistic reasons, this subsample was recruited from the last five EAs visited in each province. The same houses were revisited, and the same children and/or their caregivers interviewed. Comparison of sociodemographic variables between those who completed one 24-h recall and those who completed repeated recalls showed only two significant differences i.e., marital status and ethnic group [25] confirming similarity between the group who completed the first 24-h recall and the subsample for gender, ethnicity, area of residence, parent (mother and father) education and employment, Wealth Index, presence of hunger and body mass index of the mother. Whether the 24-h recall was less, the same, or more than the child’s usual intake, was also recorded for the total group, as well as for the two additional recalls completed for the subgroup. The 24-h recall data were analyzed using the South African Food Composition Tables (SAFCT) [30]. 

### 2.3. Application of the NCI Method in Estimation of Usual Intake of the Sample

The data obtained from the three 24-h recalls of the subsample were used to adjust the observed distributions of the single 24-h recall completed by the larger sample for the effects of random within-person variation, to establish usual intakes.

The application of the NCI-method [6] in this study focused on the estimation of usual intake of calcium, iron, zinc, thiamine, riboflavin, niacin, folate and vitamins B6, B12, A, D, E and C, as well as the estimation of proportion of intake deficiency or excessive intake. Total energy and macronutrients consumed, as well as percentage energy from fat, carbohydrates and protein, which are considered as the ratio of two dietary components consumed daily, were also estimated. As the focus of this paper is on the estimation of the risk of micronutrient deficiency or excessive intake, results for energy and macronutrients are presented in Appendix A. Application of the NCI method based on estimated intake, is also referred to as the amount-only method [6]. The advantage of this method is that it is possible to estimate usual intake distributions even if repeated dietary data are available for a relatively small sub-sample of the total population. 

Davis et al. [31] point out that a requirement for the use of the amount-only method is that for each nutrient under consideration the percentage of 24-h recalls (unweighted) with zero intake for a particular nutrient should be less than 5% within each stratum. This was the case for all nutrients in each of the three age group strata considered, with the exception of vitamin D in the 3–<6-year-olds and 6–<10-year-olds, where the percentage zero intakes were between 5% and 10%. Herrick et al. [16] explained that the amount-only model may still be applied with caution when zero intakes are between 5% and 10%.

The NCI-method steps were adapted and implemented from the outline provided by the Australian Health Survey: User’s Guide 2011–2013 [32] and Luo et al. [33] as follows:

#### 2.3.1. Step 1: Input Day-1, Day-2 and Day-3 24-h Recall Intakes

Apart from the 24-h recall on the large representative sample, two additional repeated recalls were conducted by the same interviewer on nonconsecutive days for a subset of children to ensure that observations within individuals were independent. Preliminary data adjustments were made by setting zero values to half of the minimum amount values (data must ideally contain <5% zero values). 

#### 2.3.2. Step 2: Calculate Balanced Repeated Replication Weights

In addition to the sampling weights, it is necessary to determine balanced repeated replication (BRR) weights, to obtain standard errors and confidence intervals for the mean and percentiles from the distribution of the usual intake. The BRR method [34] was calculated with a Fay coefficient of 0.3 [35]. Two pseudo primary sampling units (pseudo-PSUs) were created per stratum by randomly selecting half of each enumerator area (or cluster) in each stratum into one pseudo-PSU, and the rest in a second pseudo-PSU. Therefore, six original strata were maintained with 12 pseudo-PSUs, two per stratum. Consequently, eight sets of BRR weights were created, which is the smallest integer divisible by 4 and greater than the number of strata (which is 6), taking the original sampling weights as well as the age and gender of each child into consideration. This implies that 8 runs were executed in the process of variance estimation using the 8 sets of BRR weights. Additionally, a base run was also executed, using the original sampling weights.

#### 2.3.3. Step 3: Fit the Model and Box-Cox Transform to Near Normality

The usual intake calculations in the NCI method require a normal or near normal distribution. However, as the nutrient intake data was positively skewed a Box-Cox transformation was included in the modeling to transform the input data to normality or near normality using the following Box-Cox functions [36]:(1)$$g\left(x;\lambda \right)=\left(x^{\lambda }-1\right)/\lambda ,\;\mathrm{when}\;\lambda \neq 0,$$
(2)g(x;λ)=log(x), when λ=0, and
(3)g(x;λ)=sqrt(x), when λ=0.5.

The value of lambda (λ) associated with the Box-Cox transformation determines the strength of the transformation and is calculated as part of the overall model fitting process. Lambda = 0 reflects an extremely skewed distribution, and thus an unsuccessful transformation [11]. A *λ*-value < 0.15 reflects a larger mean bias as a result of sensitivity to the transformation applied [16]. For this research, *λ*-values associated with the first execution or base run of the macros are reported.

The covariates included in the model to represent the effect of personal characteristics were province, area of residence (formal urban, informal urban and rural areas), gender of the child [25] and whether the day the single 24-h recall was recorded in the larger group represented a usual day as perceived by respondents. These person-specific characteristics allow for individuals’ usual intake to vary between persons. To model within-person variation, two indicator variables were included as covariates to reflect whether the recall number is the second or third recall with the first recall as reference [11]. The analyses were stratified according to three age groups, namely 1–<3-year-olds, 3–<6-year-olds and 6–<10-years-olds, allowing for different transformation parameters per strata. Stratification according to age also allows for better approximations of normality for each age group to prevent highly skewed distributions [16]. 

The relationship of the covariates to the reported intakes was estimated by fitting a non-linear mixed effect model. The model is written as: (4)$$g\left(R_{ij};\lambda \right)=\beta_{0}+\sum_{k=1}^{K}\beta_{k}X_{ki}+\sum_{l=1}^{L}\beta_{l}Z_{lij}+d_{ij},$$
(5)$$d_{ij}=u_{i}+e_{ij},$$
where $R_{ij}$ denotes the recall of individual i on day j, $g\left(x;\lambda \right)=\left(x^{\lambda }-1\right)/\lambda$ is the Box-Cox transformation, $X_{ki}$ is the value of covariate *k* for person *i*,$\;Z_{lij}$ is the value of covariate *l* for person i on day j,${\;\beta }_{k}$ and $\beta_{l}$ are regression coefficients, and $d_{ij}$ is a zero-mean regression error that is further decomposed into a zero-mean person-specific effect $u_{i}$ and a zero-mean within-person error $e_{ij}$. The transformation is assumed to produce normally distributed terms $u_{i}$ and$\;e_{ij}$, which implies that $d_{ij}$ is also normally distributed. As mentioned previously, with three 24-h recalls on a subset of individuals, it is possible to disaggregate the total residual variation (the variance of $d_{ij}$) into between-person and within-person components (the variances of $u_{i}$ and $e_{ij},$ respectively) [11,33].

The NCI (amount-only) method was applied using a set of macros written in the SAS programming language. The MIXTRAN macro evaluates the effects of individual covariates on usual intake and generates parameter estimates and linear predictor values used as inputs for the DISTRIB macro (next step) [11]. The MIXTRAN macro fits a nonlinear mixed effects model to the three 24-h recalls, using the SAS NLMIXED procedure. In this application, the model is for the amount of a nutrient consumed every day (i.e., amount-only model) [6]. 

#### 2.3.4. Step 4: Simulate Usual Intakes Based on the Fitted Model

The DISTRIB macro incorporated parameter estimates and linear predictor values from the MIXTRAN macro in a Monte Carlo simulation to estimate the distribution of usual intake. The number of simulations per individual was taken as 100, as suggested in Tooze et al. [11].

#### 2.3.5. Step 5: Back-Transform to Original Scale

The simulated intake amounts were then back-transformed to the original scale, using the 9-point numerical integration approximation method, specifically the gauss-hermite quadrature method (as described in the distrib_bivariate.macro.v1.1.sas macro). The back-transformed values represent the usual intake distribution of a simulated population. This is also part of the DISTRIB macro.

#### 2.3.6. Step 6: Derive Percentiles and Proportions above/below Cut-Points

The cut-points considered in this study included the Estimated Average Requirement (EAR; daily intake value of a nutrient that is estimated to meet the nutrient requirement of half the healthy individuals in a life stage and gender group; intake below the EAR denotes risk of deficiency), the Tolerable Upper Intake Level (UL; the highest level of daily nutrient intake that is likely to pose no risk of adverse health effects to almost all individuals in the general population), the Acceptable Macronutrient Distribution Ranges (AMDRs) for carbohydrate, protein and fat, as well as the Estimated Energy Requirement (EER) [37].

After executing the DISTRIB macro, the Percentiles Survey macro was used to read the back-transformed usual intake values calculated in the Monte Carlo simulation, calculate the percentiles of usual intake, as well as to calculate the percentages below, within or above given EAR, UL, EER and AMDR cut-off values. Sample weights and BRR weights were considered to ensure the results are representative of the population.

The SAS programs and the respective macros used are available from the web page of the US Department of Health and Human Services [38].

### 2.4. Statistics

For usual intakes, the mean intake, standard error of the mean intake, median intake, standard error of the median intake, Box-Cox parameters, between-person variance, within-person variance, the ratio of within-person variance to between-person variance and the coefficient of variation, which is the ratio of the standard error and the mean expressed as a percentage, are reported. Additionally, proportions below EARs, above ULs (micronutrients); below the EER (energy); and above, within or below AMDRs (macronutrients) were calculated. All these calculations were performed using the NCI method as described above.

For Day-1 intakes, weighted means with standard errors, medians with standard errors, as well as proportions below EARs, above ULs (micronutrients); below the EER (energy); and above, within or below AMDRs (macronutrients) were calculated for each age group. Mean and median intake values were compared between Day-1 and usual intake for all nutrients using the t-test (means) and the Kruskal–Wallis test (to compare the locations of two populations, specifically the medians) after the INDIVINT macro was executed. The INDIVINT macro uses parameter estimates from the MIXTRAN macro to predict individual nutrient intake for further use in disease models [24]. The percentage difference between mean and median usual and Day-1 intakes was calculated for each nutrient within each age group using the formula as used by Davis et al. [31] where
(6)$$\%\mathrm{Difference}\;\mathrm{in}\;\mathrm{intake}=100\frac{\left(\mathrm{usual}\;\mathrm{intake}-\mathrm{Day}1\;\mathrm{intake}\right)}{\mathrm{Day}1\;\mathrm{intake}}.$$

We considered a percentage difference of >10% as a concern. 

The proportions (as a percentage) <EAR, <EER and <AMDRs for micronutrients, energy intake and macronutrients respectively were compared by calculating the difference between the proportions, using EAR as an example, as follows:(7)%Difference in risk=%(usual intake<EAR)−%(Day1 intake<EAR).

This difference provides a descriptive perspective on the magnitude of the difference between the two estimates of risk for each nutrient.

Simulated distributions and cumulative distributions of the usual intakes were visually compared to the corresponding distributions of Day-1 intakes. DRI cut-points were also included in these figures.

The top 5 food contributors to micronutrient intake were determined to formulate food-based explanations for potential outliers in the Day-1 data set. The percentage contribution of food items to the total intake of each of the investigated micronutrients and percentage eaters of these food items were calculated for these purposes. Interpretations focus on the top 5 food sources that were consumed by less than 10% within each age group for each micronutrient. These calculations were not done with the usual intake data set because of the complexity of the NCI method where 100 simulations are generated for each individual. All analyses were conducted using SAS Version 9.4, SAS for Windows (SAS Institute, Cary, NC, USA).

## 3. Results

The macro- and micronutrient results have been published and discussed comprehensively in Steyn et al. [25] (energy and macro-nutrients) and Senekal et al. [2] (micronutrients). 

Results in the current study are used to demonstrate the importance of removal of within person variation and not to reflect on adequacy per se, as published in Steyn et al. [25] and Senekal et al. [2]. Results are presented in four series, with each series including results for the three age groups separately: (1) three minerals (calcium, iron and zinc); (2) six B-vitamins (thiamine, riboflavin, niacin, folate and vitamins B6 and B12); (3) three fat-soluble vitamins (A, D and E) and vitamin C; and (4) energy and macronutrients (included as Appendix A). Each series includes four tables and a composite figure to illustrate the various concepts linked to the NCI method to remove intra-individual variation from single 24-h recall data to estimate usual intake. The first table in each series presents the mean (standard error (SE) of the mean), median (SE of the median) and percentage difference between usual and Day-1 intake for all variables. The second table shows the values for the within-person and between-person variances, the ratio of within-person relative to the between-person variance and the coefficient of variation (CV), which is the ratio of the standard error of the mean and the mean of usual intake, expressed as a percentage. The third table compares the percentage of the total sample above or below DRI cut-points for Day-1 and usual intake results. All analyses incorporate sample weights, the complex survey design, and in the case of usual intakes, BRR weights. The fourth table lists the top foods contributing to the nutrient intakes.

The figure in each series includes a set of sub-figures to illustrate the distribution of the simulated, back-transformed values (usual intake) and the distribution of Day-1 intakes for the nutrients within the context of their DRIs. All results are presented by age group (1–<3-year-olds, 3–<6-year-olds and 6–<10-year-olds).

### 3.1. Minerals

Mean usual calcium, iron and zinc intakes were similar and median intakes significantly higher when compared to Day-1 values in all three age groups. For calcium the percentage difference in medians between the two methods was above 10%, while it was below 10% for zinc and iron in all three age groups (Table 1). 

The ratio of the within-person to between-person variance was well below 10 for all three minerals across the age groups (Table 2). The λ-values of all three nutrients reflect successful transformations (λ ≥ 0.15) [16]. The coefficient of variation decreased with age for all three minerals.

The percentage children with a calcium intake below the EAR (risk of deficicency) was underestimated when using Day-1 intake [%(usual intake < EAR) minus %(Day-1 intake < EAR)] by 4% for the 1–<3-year-olds, 7.5% for 3–<6-year-olds and 3.5% for 6–<10-year-olds (Table 3; Figure 1). The two methods resulted in similar estimations of risk of iron deficiency, with the percentage difference in estimates being less than 3% in all three age groups. The two methods also resulted in a similar estimate of risk of zinc deficiency in the youngest age group, but Day-1 intake overestimated risk of zinc deficiency with 8.4% and 7.5% in the two older age groups respectively. Estimation of intake above the UL (risk of excess) [%(usual intake > UL) minus %(Day-1 intake > UL)] was the same for calcium and iron for Day-1 and usual intakes across the three age groups (no child had an intake above the UL for these minerals according to both methods). Estimations for zinc intake above the UL were similar for the two younger age groups, but the estimation based on Day-1 intake was 8.5% higher than the estimate based on usual intake in the oldest age group (Table 3, Figure 1). 

Figure 1 presents the density functions of Day-1 and usual intakes within the context of DRI cut-points to further illustrate the results presented in Table 1, Table 2 and Table 3, with the difference in risk of deficiency and over-exposure between the two data sets clearly illustrated. Figure 1 demonstrates how the spread of the usual intake density function becomes narrower when the within-person variance is addressed. The narrowing of the spread of the usual intake density function is evident for all three minerals, for all three age groups, but is specifically prominent in the 3–<6-year-olds. 

Table 4 provides a list of the top 5 food items that contributed to calcium, iron and zinc intakes, by age group, using Day-1 intake data. The top food for each of the three minerals was the same in all three age groups, whole milk for calcium and maize porridge for iron and zinc. These items were consumed by 44% or more of the children in each age group. Note that pilchards were consumed by a small percentage of the youngest age group (8%), but made a material contribution to calcium intake (6%). The same is true for organ meat intake in the 3–<6-year-old group where 9% consumed organ meat which contributed 4% to total iron intake.

### 3.2. Vitamins (Excluding B Vitamins)

Mean usual intakes were similar for vitamins A, C, D and E, while median usual intakes were significantly higher when compared to Day-1 values in all three age groups (Table 5, Figure 2). For vitamin A, the percentage difference in medians between the two methods was 44.0%, 44.9% and 27.0% from the youngest to oldest age group; for vitamin D it was 100%, 91.7% and 45%; for vitamin E it was 17.7%, 25% and 23.2% and for vitamin C it was 22.9%, 55.1% and 36.3%, respectively (Table 5, Figure 2).

The ratio of the within-person to between-person variance was well below 10 for all four vitamins across the age groups. The λ-value for vitamin A is zero, reflecting an unsuccessful transformation and thus an extremely skewed distribution. The coefficient of variation decreased with age for all four vitamins (Table 6).

Comparison of the estimated percentage children with a vitamin A intake below the EAR (risk of deficiency) between usual intake and Day-1 intake, shows that Day-1 overestimated deficiency risk [%(usual intake < EAR) minus %(Day-1 intake <EAR)] in 1–<3-year-olds with 14.9%, in 3–<6-year-olds with 24.1% and 6–<10-year-olds with 17.3% (Table 7, Figure 2). A similar scenario is evident for vitamin C, where Day-1 results overestimated deficiency risk with 13.9% in 1–<3-year-olds, 30.1% in 3–<6-year-olds and 15.0% in 6–<10-year-olds. Day-1 intake results also overestimated vitamin E intake below the EAR when compared to usual intake results with 18.0% in 1–<3-year-olds, 19.7% in 3–<6-year-olds and 16.4% in 6–<10-year-olds. Risk of vitamin D deficiency was close to 100% based on both methods (Table 7, Figure 2). Estimates of an intake above the UL (risk of excess) was the same according to both methods for vitamins C, D and E across the three age groups (no child had an intake above the UL for any one of these nutrients according to both methods). Day-1 intake results underestimated risk of excess [%(usual intake > UL) minus %(Day-1 intake > UL)] of vitamin A with 9.7% in 1–<3-year-olds, 2.8% in 3–<6-year-olds and 1.7% in 6–<10-year-olds (Table 7, Figure 2).

Figure 2 presents the density functions of Day-1 and usual intakes within the context of DRI cut-points to further illustrate the results presented in Table 5, Table 6 and Table 7, with the difference in risk of deficiency and over-exposure being clearly evident. The usual intake density function for vitamin A shifts to the right of the Day-1 intake density function, which is extremely skew, especially for age groups 3–<6-years and 6–<10-years, with the tails becoming narrower with the removal of the within-person variance. However, despite the adjustment, a long tail for vitamin A is still evident in all three age groups, reflecting the lambda value that is close to zero. Similar trends are evident for vitamins C, D and E, but not as extreme, as the lambda values reflect successful transformations (λ = 0.14 for vitamin E, and λ > 0.15 for vitamins C and D) (Figure 2, Table 6).

The top 5 food items that contributed to the intake of vitamins A, C, D and E are presented in Table 8. Organ meat was the top source of vitamin A in the two older age groups, providing 32% and 27% respectively towards total intake, although eaten by only 9% in each age group. Organ meat was third in the top five foods contributing to vitamin A intake in the youngest age group (11% contribution, 5% eaters). One further top five food contributor to vitamin A intake, namely carotene-rich vegetables other than green leafy vegetables, was consumed by less than 10% of children in the 1–<3-year-olds (14% contribution, 9% eaters) and 6–<10-year-olds (10% contribution, 9% eaters). 

In the youngest age group, none of the top 5 food items contributing to vitamin C intake was consumed by less than 10% of the group. Fruit juice was the top contributor to vitamin C intake in the two older age groups (18% and 23% respectively), although only consumed by 7% in each of these two groups. Fresh vitamin C-rich fruit contributed second most to vitamin C intake in the oldest age group (17%), although eaten by 9% only.

Pilchards featured in the top five food contributors to vitamin D intake (16%, 25% and 32% contributions respectively; 6%, 8% and 13% eaters respectively). Fat cakes (bread dough fried in oil) contributed fourth most to vitamin D intake in the 6–<10-year-olds (4%), although only eaten by 7%. The only top five contributor to vitamin E intake that was eaten by less than 10%, was fat cakes in the 6–<10-year-olds (7% contribution, 7% eaters).

### 3.3. B-Vitamins

Mean usual intakes of all B-vitamins were similar, while median usual intakes were significantly higher when compared to Day-1 values in all three age groups, with the exception of thiamine in the 1–<3-year-olds where there was no difference. The percentage difference in median intakes between the two methods was less than 10% for thiamine and riboflavin in the youngest and oldest age groups, but above 10% in the 3–<6-year-olds. For niacin and vitamin B6 the percentage difference in median intakes was below 10% in all three age groups. For vitamin B12, the percentage difference in median intakes was 54.6%, 123.1% and 105.8% from youngest to oldest group respectively; this was less than 10% for the youngest and oldest age groups for folate, but 18% for the 6–<10-year-olds (Table 9, Figure 3).

The ratio of the within-person to between-person variance was well below 10 for all the B-vitamins across the age groups. The λ-values of thiamine, niacin, riboflavin and vitamin B6 reflect successful transformation (λ ≥ 0.15) (Table 10). The λ-values of vitamin B12 (0.13) and folate (0.7) reflect presence of skewness of the adjusted data. The coefficient of variation was the highest in the 1–<3-year-olds and the lowest in the 3–<6-year-olds for all the B vitamins. 

Comparison of the percentage children with a thiamine intake below the EAR shows that Day-1 intake results overestimated deficiency risk [%(Usual intake < EAR) minus %(Day-1 intake < EAR)], but with less than 4% in the youngest and oldest age group and with 6.3%, in 3–<6-year-olds (Table 11, Figure 3). For niacin Day-1 intake overestimated risk of deficiency with 10.6% in 1–<3-year-olds, with the difference being less than 6% in the two older groups. Day-1 intake results further overestimated riboflavin intake below the EAR with more than 10% in all three age groups. For vitamin B6 the overestimation was less than 3% in all three age groups, while the difference in overestimation of risk for vitamin B12 and folate deficiency by Day-1 intakes was above 20% for vitamin B12 and above 12% for folate across the age groups (Table 11, Figure 3). Day-1 intake results overestimated risk of excess [%(usual intake > UL) minus %(Day-1 intake > UL)] with more than 50% for niacin, but less than 6% for folate across the three age groups. No child had an intake above the UL for vitamin B6 based on Day-1 or usual intake results (Table 11, Figure 3).

Figure 3 presents the density functions of Day-1 and usual intakes within the context of DRI cut-points to further illustrate the results presented in Table 9, Table 10 and Table 11, with the difference in risk of deficiency and over-exposure being clearly evident. Some narrowing of the tails for thiamine and riboflavin intake distribution functions is evident in the two older age groups with the removal of the within-person variance, but the intake density function is very similar between Day-1 and usual intake estimates in the youngest age group. Clear narrowing of the tails is evident for niacin and riboflavin in all three age groups and for vitamin B12 in the two younger age groups. For folate there is a small shift in the intake density function to the right with the adjustment and long tails are still evident in all three age groups, reflecting the lambda value of 0.07 (Figure 3, Table 9, Table 10 and Table 11). 

All top food items that contributed to thiamine, niacin and vitamin B6 intakes respectively were consumed by 10% or more of the children, with maize porridge, white bread and/or brown bread and/or high/low fiber fortified cereal being in the top 5 in all three age groups (Table 12). A similar profile is evident for riboflavin, with the exception of the exclusion of bread as a top 5 source in all three age groups and inclusion of organ meat in the top 5 in the 3–<6-year-olds and 6–<10-year-olds. Organ meat was consumed by 9% of children in each of these age groups, with the contributions to riboflavin intake being 10% and 8% respectively (Table 12). 

As far as vitamin B12 is concerned, two food items in the top 5 sources, organ meat and pilchards, were consumed by less than 10% of children, with the exception of pilchards that was consumed by more than 10% in the 6–<9-year-olds. Organ meat and pilchards were consumed by 5% and 6% respectively in 1–<3-year-olds, supplying 11% and 32% respectively of vitamin B12 intake. Nine percent of 3–<6-year-old children consumed organ meat and 8% pilchards, contributing 42% and 24% respectively to vitamin B12 intake. Nine percent of 6–<10-year-olds consumed pilchards, which contributed 30% towards their vitamin B12 intake. Maize porridge, white bread and/or brown bread and/or high/low fiber fortified cereal were in the top 5 sources of folate and were consumed by 10% or more children in all three age groups. Organ meat was also a top 5 source of folate, but was consumed by 5% in 1–<3-year-olds (contribution: 5%); 9% in 3–<6-year-olds (contribution: 9%) and 9% in 6–<10-year-olds (contribution: 7%) (Table 12). 

## 4. Discussion

Estimation of risk of micronutrient deficiency using data from a single unadjusted 24-h recall (Day-1) resulted in material overestimation of deficiency risk for vitamins B12, A, C, D and E in 1–<10-year-old children (example data set) when compared to usual intake where within-person variance was removed using the NCI method. Micronutrients for which food sources were episodically consumed but contributed materially towards total intake of a particular nutrient, seemed to be particularly prone to overestimation of risk of deficiency when intra-individual variance was not removed. Overestimation of risk of excess (intake above the UL) by the single unadjusted 24-h recall compared to usual intake was present, but less prominent. 

As described in the methods section, the NCI method as applied in this research involves complex processing of 24-h recall data with a second and third day repeat for a small representative sub-sample; determination of balanced repeated replication (BRR) weights; the fit of a non-linear mixed effect model for each nutrient under discussion with the dependent variable the Box-Cox transformed value of the nutrient; simulation to estimate the distribution of usual intake; back-transformation to the original scale that represents the usual intake distribution; and finally, the calculation of means, standard errors, percentiles of usual intake (not reported in tables), as well as the percentages below, within or above given cut-off values [38]. As indicated by Herrick et al. [16], we found that adapting and running these macros is complex and time consuming.

Herrick et al. [16] emphasize that the distribution of nutrients that are consumed daily may be skewed and need to be transformed for data analyses to ensure that all key NCI model assumptions are met when the amount-only method is used. Davis et al. [31] advise that to meet these assumptions for each nutrient under consideration the percentage of 24-h recalls (unweighted) with zero intake for a particular nutrient should be less than 5% within each stratum. This was the case for all nutrients in each of the three age group strata in the example data set considered, except for vitamin D in the 3–<6-year-olds and 6–<10-year-olds, where the percentage zero intakes were between 5% and 10%. Herrick et al. [16] indicated that the amount-only model may still be applied with caution when zero intakes are between 5% and 10%. As all our data were non-normally distributed, it was transformed, with results showing that the transformation was successful (λ ≥ 0.15) for most of the investigated micronutrients. The exceptions were vitamin A (λ = 0.0) and folate (λ = 0.07), while transformation of vitamin B12 (λ = 0.13) and vitamin E (λ = 0.14) were close to successful. Other studies also obtained similar low lambda values for vitamin A [11,32] and concluded that usual intake values obtained for vitamin A should be interpreted with caution. 

According to Davis et al. [31], when the within-person to between-person variance ratio is less than 10, a limited number of records included in the analysis is sufficient for estimation of usual intake. The within-person variation was greater than the between-person variation for all nutrients we investigated, except for thiamine and iron in the 1–<3-year-olds where the ratio was 1. Furthermore, the ratio was well below 10 for all but one micronutrient in all three age groups in the example data set, indicating that the NCI method was successfully applied to account for variations in the estimation of usual intake of children in the study. The exception was vitamin D in 3–<6-year-olds where the between-person variance in vitamin D intake was extremely low (0.07), resulting in a ratio of 271. However, average results for this vitamin need to be interpreted with caution as vitamin D content of 16% of the food items in the MRC South Food Composition tables has not yet been established (Personal communication: J Chetty, SAMRC).

In line with findings of others who compared various methods for estimating usual dietary intake, the means for all nutrients we investigated were similar between the single unadjusted 24-h recall and NCI methods. As could be expected from the non-normal distribution of the single unadjusted 24-h recall data, median usual intakes for all but one micronutrient in one age group in the example data set were significantly higher than medians from the single unadjusted 24-h recall. The exception was thiamine in the 1–<3-year-olds where there was no difference. To gain insights into whether these differences are material in terms of dietary intervention decision making, we considered the percentage difference between the medians, with a difference of 10% or more considered to be material. We also propose comparison of estimations of risk of deficiency and excess, as Herrick et al. [16] state that compression of intake distributions towards the mean is most apparent when examining the proportion of the population below the EAR or above the UL [16]. 

The percentage difference between medians for children in the data set according to the two methods was found to be above 10% for vitamins A, C, D, E and B12. For vitamin B12, the differences were substantial and ranged between 54.6% and 123.1%; for vitamin D these ranged between 45% and 100%; for vitamin A between 27% and 44.9%; for vitamin C between 22.9% and 55.1% and for vitamin E between 17.7% and 25% across the three age groups. The percentage difference in medians was also more than 10% for riboflavin, vitamin B6 and folate in 3–<6-year-olds. For the remaining micronutrients, the percentage difference was less than 10% across age groups. As a result, the single unadjusted 24-h recall intake when compared to usual intake overestimated risk of vitamin B12 deficiency (% < EAR) with more than 20% in all three age groups; of vitamins A and C deficiency with more than 20% in at least one age group; and of vitamin E, riboflavin and folate deficiency with between 10% and 20% in all three age groups, clearly reflecting compression of the left tail of the distribution towards the mean for these nutrients with the application of the NCI method. Piernas et al. [9] also found that a single 24 h recall overestimated risk of deficiency, compared to usual intake calculated using the NCI method with a 2nd recall in a subsample as small as 9% of children, of zinc (6% and 1% respectively) and iron (7% and 1%) in 24–<48-month-old children. However, they indicate that this level of difference would not have public health implications. 

Estimations of intake above the UL (risk of excess) were prominent for vitamin A and zinc in 1–<3-year-olds. The single unadjusted 24-h recall underestimated risk of excess vitamin A when compared to usual intake (27.8% versus 37.5% respectively). This was 18.7% versus 21% respectively for 3–<6-year-olds and 12.1% versus 13.7% respectively for 6–<10-year-olds. For zinc, estimations of excess were similar between the two methods for the two younger age groups (1–<2-year-olds %: 35.0% versus 35.3%; 3–<6-year-olds: 21.6% versus 20.9% respectively and 13.2% versus 4.7% respectively in 6–<10-year-olds). Piernas et al. [9] also found that risk of excess zinc intake was underestimated by a single 24-h recall compared to usual intake (48% versus 73% respectively), but no difference between the methods was found for iron.

The differences found in the deficiency risk estimations for vitamins A, C and B12 between the two methods may be linked to occasional consumption of foods which are good sources of these nutrients that may not be captured by a single 24-h recall. Results of the single unadjusted 24-h on the top food contributors to vitamins A, C and B12 illustrate this point. Organ meat, consumed by 5% of 1–<3-year-olds, 9% of 3–<6-year-olds and 9% of 6–<10-year-olds, contributed 11%, 42% and 30% of vitamin B12 respectively and 11%, 32% and 27% to vitamin A intake. Pilchards were consumed by 6% of 1–<3-year-olds, 8% of 3–<6-year-olds and 13% of 6–<10-year-olds, contributing 32%, 24% and 36% respectively to vitamin B12 intake and 27%, 22% and 22% respectively to vitamin A intake. For vitamin C, the item consumed by only 6% of 1–<3-year-olds, 7% of 3–<6-year-olds and 7% of 6–<10-year-olds, but contributed 14%, 18% and 23% respectively to total intake, is fruit juice. The importance of adjustment for the effect of foods that are only consumed occasionally is clearly illustrated by these results.

The finding that estimations of risk of deficiency were more aligned between the two dietary methods for micronutrients (other than vitamin A) included in the South African fortification mix, namely zinc, iron, thiamin, riboflavin, niacin, and vitamin B6, and that overestimation of risk of deficiency was less than 10% for all these nutrients across age groups, could possibly be explained by the fact that these fortified foods were consumed daily in all three age groups [2]. This is further supported by our findings from the current study that either maize or bread or both were included in the top three food sources for all seven of these nutrients in all three age groups. Herrick et al. [16] also mentioned that nutrients provided by ubiquitously consumed foods may be less prone to intra-person variance than nutrients provided largely by foods consumed episodically. This suggests that a single unadjusted 24-h recall could provide a reasonable reflection of the intake of micronutrients supplied by moderate to good sources that are consumed daily. This is also true for energy and macronutrient intake as reflected by our results and that of others [9,16,19,24]. 

The differences we found in deficiency risk estimation for vitamins A, C and B12 in the data set may have serious implications for decisions made regarding fortification. Consider the example of vitamin B12 intake in 3–<6-year-olds where deficiency risk was estimated to be 35% when using the single unadjusted 24-h recall method versus 0% based on the usual intake method. When a third of a population is at risk of deficiency, it may be considered a potential public health problem and prompt a decision to intervene. For example, in South Africa the decision may well be to include vitamin B12 in the existing mandatory fortification mix. The same is true for vitamin C, where risk of deficiency drops from 21.3%, 39.1% and 40.3% from the youngest to oldest age group (single, unadjusted 24-h recall), to 7.4%, 9.0% and 25.3% (usual intake). Although the risk of vitamin C deficiency in the 6–<10-year-olds could be a real concern, the intervention decision based on the single unadjusted 24-h recall estimation may be to fortify, while the usual intake estimates may prompt a decision to promote an increase in fruit and vegetable intake based on the finding that fruit and vegetable intake in the study sample was only approximately 110 g of the recommended 400 g/day in all three age groups [2]. Conversely, bearing in mind that vitamin A results were still skewed after transformation, the estimate of a deficiency risk of 24.5% in 3–<6-year-olds based on the single unadjusted 24-h recall data (versus 0.4% based on usual intake) may erroneously result in a conclusion that the amount of vitamin A included in the mandatory fortification mix in South Africa is not sufficient to alleviate risk of deficiency in a quarter of children in this age group.

Limitations of this research include that the NCI method was originally developed to be applied to the US Department of Agriculture’s National Health and Nutrition Examination Surveys (NHANES), with its unique PSU, strata system and weighting factors [9]. The example data set has a different PSU scheme due to the unique strata and cluster design. Moreover, we had to design a unique set of Balanced Repeated Replication weights. As mentioned, ideally each participant should supply dietary data for at least two days for application of the NCI method [9,16] in the example data set, but additional recalls were only available for 11% of the total sample. Results thus need to be interpreted bearing in mind that the approximation of the within-person variance may have been affected by the smaller sample size of the second and third recalls. Further limitations are that the distribution of vitamin A intake remained skew after transformation and the high variance ratio for vitamin D in the middle age group. Souverein et al. [23] caution that highly asymmetrical distributions may lead to implausible estimates. Finally, as there is no gold standard for assessment of dietary intake estimated values could not be confirmed against true usual intake [24].

Important strengths of our study include the large sample size in the data set, which allowed a meaningful illustration of the importance of the removal of large within-person variability in dietary intake surveys. Additionally, dietary intake in the example data set were collected using a standardized protocol to minimize the bias inherent to the self-reported nature of 24-h recall as much as possible. 

## 5. Conclusions

Evidence-based intervention decisions for ensuring optimal micronutrient status of populations should be based on best evidence on presence of deficiency or excess in particular target populations. Although this evidence should ideally include biological risk indicators of micronutrients, such information is often not available. As a result, policymakers must revert to dietary intake risk data for guidance. Ideally, these data should be from nationally representative dietary surveys on usual intake [3,5]. As mentioned in the introduction, a single 24-h recall is most frequently used for these purposes [14,15,16] but is a poor estimator of long-term usual intake [10,14,15,16], as it only provides an estimate of the between-person variance.

Based on our example of application of the NCI method to adjust single 24-h recall data for large within-person variability, using two additional recalls in a sub-sample, we conclude that this adjustment makes a material difference in evidence-based estimation of risk of micronutrient deficiency/excess. In our example, micronutrients that were specifically affected were vitamins A, C, D, E and B12, with the proportion of the target population identified as being at risk of deficiency being lower after adjustment. The possibility that the unadjusted 24-h recall may also overestimate risk of excess is illustrated by the results for vitamin A and zinc in our example. We further conclude that episodically consumed foods that are rich sources of particular micronutrients may contribute specifically to overestimation of deficiency risk when intra-individual variance is not removed. 

This research clearly illustrates that when compared to estimations based on a single unadjusted 24-h recall, usual intake estimates based on the application of the NCI method would lead to more appropriate conclusions on risk of deficiency/excess in a particular population. This is key to appropriate public health nutrition intervention planning. Researchers seeking to evaluate dietary adequacy in relation to reference standards should thus make use of usual intake methods. 

Having conducted this research, we concur with the notion put forward by Luo et al. [33] that “Although the NCI method is very flexible and powerful, it has a fairly steep learning curve. Applying the method requires understanding the theoretical basis for analyzing dietary intake data, mastery of multiple SAS macros, and a high proficiency in SAS coding to enable data-set-specific modifications”. However, this method is a good option for consideration in situations where collection of repeated records is not feasible. The contribution to evidence-based public health nutrition intervention is the reward.

## Figures and Tables

**Figure 1 nutrients-14-00285-f001:**
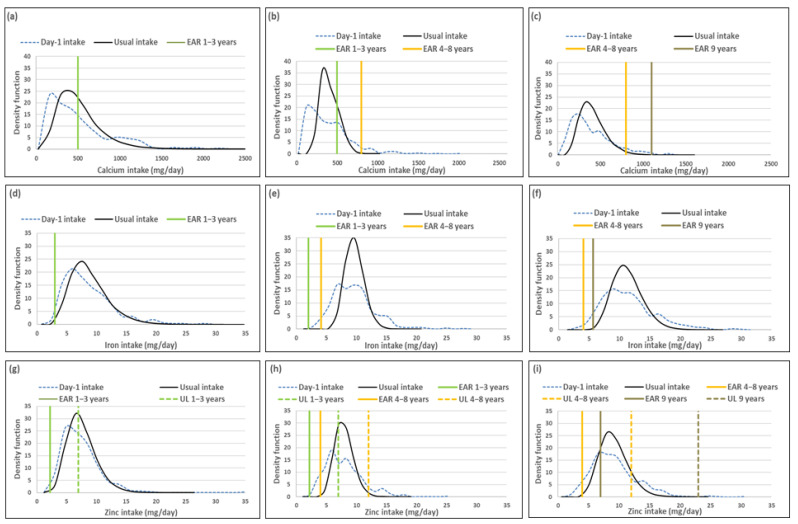
Distributions of the simulated, back-transformed values (usual intake) and the distribution of Day-1 intakes within the context of DRI-values of calcium, iron and zinc, by age group. (**a**): Age 1–<3 years: Calcium intake; (**b**): Age 3–<6 years: Calcium intake; (**c**): Age 6–<10 years: Calcium intake; (**d**): Age 1–<3 years: Iron intake; (**e**): Age 3–<6 years: Iron intake; (**f**): Age 6–<10 years: Iron intake; (**g**): Age 1–<3 years: Zinc intake; (**h**): Age 3–<6 years: Zinc intake; (**i**): Age 6–<10 years: Zinc intake.

**Figure 2 nutrients-14-00285-f002:**
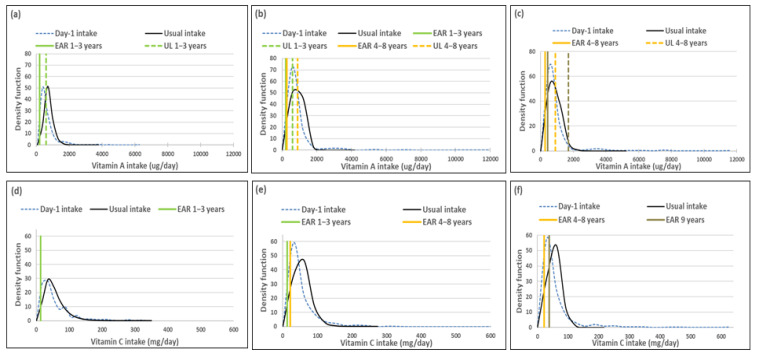
Distributions of the simulated, back-transformed values (usual intake) and the distribution of Day-1 intakes within the context of DRI-values of vitamins A, C, D and E, by age group. (**a**): Age 1–<3 years: Vitamin A intake; (**b**): Age 3–<6 years: Vitamin A intake; (**c**): Age 6–<10 years: Vitamin A intake; (**d**): Age 1–<3 years: Vitamin C intake; (**e**): Age 3–<6 years: Vitamin C intake; (**f**): Age 6–<10 years: Vitamin C intake; (**g**): Age 1–<3 years: Vitamin D intake; (**h**): Age 3–<6 years: Vitamin D intake; (**i**): Age 6–<10 years: Vitamin D intake; (**j**): Age 1–<3 years: Vitamin E intake; (**k**): Age 3–<6 years: Vitamin E intake; (**l**): Age 6–<10 years: Vitamin E intake.

**Figure 3 nutrients-14-00285-f003:**
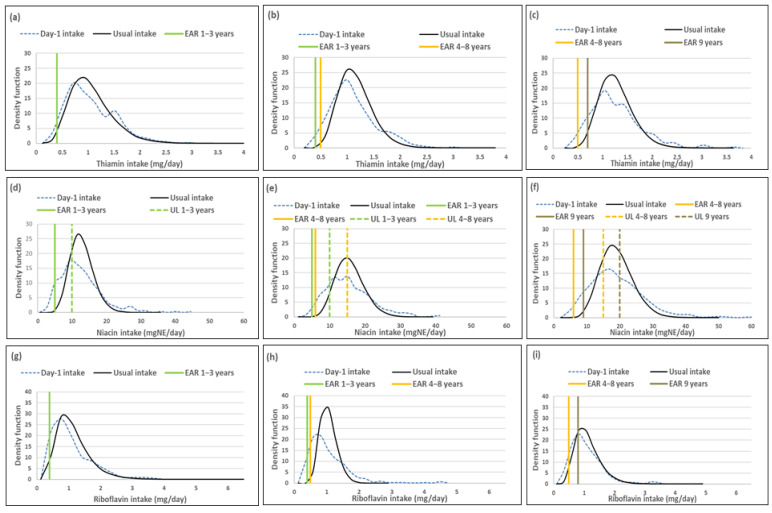
Distributions of the simulated, back-transformed values (usual intake) and the distribution of Day-1 intakes within the context of DRI-values of thiamin, niacin, riboflavin, vitamin B6, vitamin B12 and folate, by age group. (**a**): Age 1–<3 years: Thiamin intake; (**b**): Age 3–<6 years: Thiamin intake; (**c**): Age 6–<10 years: Thiamin intake; (**d**): Age 1–<3 years: Niacin intake; (**e**): Age 3–<6 years: Niacin intake; (**f**): Age 6–<10 years: Niacin intake; (**g**): Age 1–<3 years: Riboflavin intake; (**h**): Age 3–<6 years: Riboflavin intake; (**i**): Age 6–<10 years: Riboflavin intake; (**j**): Age 1–<3 years: Vitamin B6 intake; (**k**): Age 3–<6 years: Vitamin B6 intake; (**l**): Age 6–<10 years: Vitamin B6 intake; (**m**): Age 1–<3 years: Vitamin B12 intake; (**n**): Age 3–<6 years: Vitamin B12 intake; (**o**): Age 6–<10 years: Vitamin B12 intake; (**p**): Age 1–<3 years: Folate intake; (**q**): Age 3–<6 years: Folate intake; (**r**): Age 6–<10 years: Folate intake.

**Table 1 nutrients-14-00285-t001:** Mean (SE of the mean), median (SE of the median) and percentage difference (usual intake versus Day-1 intake) for calcium, iron and zinc intake, by age group.

		Age 1–<3 Years *n* = 333			Age 3–<6 Years *n* = 514			Age 6–<10 Years *n* = 479		
Nutrient		Usual Intake	Day-1 Intake	% Diff	Usual Intake	Day-1 Intake	% Diff	Usual Intake	Day-1 Intake	% Diff
Calcium (mg/day)	Mean (SE)	423.3 (45.3)	424.5 (19.1)	−0.3	350.9 (6.4)	348.5 (19.2)	0.7	351.8 (13.6)	352.2 (13.5)	−0.1
	Median (SE)	378.0 (39.2)	339.0 (19.9)	11.5 ***	329.2 (4.4)	288.3 (22.9)	14.2 ***	331.3 (15.0)	299.9 (15.0)	10.5 ***
Iron (mg/day)	Mean (SE)	7.8 (0.5)	7.7 (0.3)	1.3	8.9 (0.1)	8.9 (0.3)	0.0	10.6 (0.1)	10.6 (0.2)	0.0
	Median (SE)	7.3 (0.4)	7.2 (0.3)	1.4 **	8.8 (0.1)	8.6 (0.3)	2.3 **	10.3 (0.1)	9.7 (0.2)	6.2 **
Zinc (mg/day)	Mean (SE)	6.5 (0.4)	6.4 (0.2)	1.6	7.3 (0.1)	7.3 (0.2)	0.0	8.5 (0.2)	8.5 (0.2)	0.0
	Median (SE)	6.2 (0.4)	6.0 (0.2)	3.3 *	7.1 (0.1)	6.8 (0.3)	4.4 **	8.3 (0.1)	7.9 (0.2)	5.1 *

Day-1 intake: The reported 24-h recall on the first visit for the total group. Usual intake: Long-term daily average intake as calculated using the NCI amount-only method [6]. EAR = Estimated Average Requirement [37]. Percentage difference of the usual intake relative to Day-1 intake was computed for means and medians as follows: % Diff = 100 × (Usual intake − Day-1 intake)/(Day-1 intake). There were no significant differences between the means of Day-1 intake and usual intake, *t*-test, *p*-values > 0.05. * *p* < 0.05; ** *p* < 0.01; *** *p* < 0.001: Significant difference between locations of Day-1 and usual intakes, Kruskal–Wallis test.

**Table 2 nutrients-14-00285-t002:** Within-person and between-person variance, ratio of these two parameters and the coefficient of variation for usual intake of calcium, iron and zinc, by age group.

Nutrient (Box-Cox TP)	Age: 1–<3 Years				Age: 3–<6 Years				Age: 6–<10 Years			
	Var_e	Var_u	Ratio	CV (%)	Var_e	Var_u	Ratio	CV (%)	Var_e	Var_u	Ratio	CV (%)
Calcium (λ = 0.24)	7.12	4.48	1.58	10.7	7.30	1.13	6.46	1.8	4.67	2.51	1.86	3.9
Iron (λ = 0.18)	0.24	0.24	1.00	7.1	0.28	0.08	3.50	1.4	0.22	0.13	1.69	1.1
Zinc (λ = 0.20)	0.28	0.19	1.47	6.9	0.31	0.11	2.82	2.0	0.29	0.13	2.23	1.8

A lambda value < 0.15 reflects a larger mean bias as a result of sensitivity to the transformation applied [16]. Lambda-values associated with the first execution or base run of the macros are reported. Var_e: Within-person variance; Var_u: Between person variance; Ratio = Var_e/Var_u, the ratio of within-person to between-person variance. CV: Coefficient of variation of the mean of usual intake, where CV = 100 × (standard error of the mean)/mean. TP: Transformation parameter.

**Table 3 nutrients-14-00285-t003:** Comparison of percentage below EAR and above UL cut-points for risk of calcium, iron and zinc deficiency and excess respectively, by age group.

		Day-1 Intake		Usual Intake		Difference ^1^ for % < EAR and (% > UL)
	Age Group	% < EAR (95% CI)	% > UL (95% CI)	% < EAR (95% CI)	% > UL (95% CI)	
Calcium (mg/day) EAR-UL: 1–3 years = 500–2500 mg; 4–8 years = 800–2500 mg; 9–<10 years = 1100–3000 mg	1–<3 years (*n* = 333)	66.2 (59.9–72.6)	0.0 (-)	70.2 (51.1–89.3)	0.0 (-)	4.0% (0.0%)
	3–<6 years (*n* = 514)	87.3 (83.3–91.2)	0.0 (-)	94.8 (91.5–98.2)	0.0 (-)	7.5% (0.0%)
	6–<10 years (*n* = 479)	95.9 (93.5–98.2)	0.0 (-)	99.4 (98.3–100.0)	0.0 (-)	3.5% (0.0%)
Iron (mg/day) EAR-UL: 1–3 years = 3–40 mg; 4–8 years = 4.1–40 mg; Male:9–<10 years = 5.9–40 mg; Female:9–<10 years = 5.7–40 mg	1–<3 years (*n* = 333)	3.4 (1.0–5.7)	0.0 (-)	1.0 (0.0–3.2)	0.0 (-)	−2.4% (0.0%)
	3–<6 years (*n* = 514)	2.7 (1.0–4.3)	0.0 (-)	0.01 (0.0–0.1)	0.0 (-)	−2.7% (0.0%)
	6–<10 years (*n* = 479)	2.5 (0.9–4.2)	0.0 (-)	0.3 (0.0–0.8)	0.0 (-)	−2.2% (0.0%)
Zinc (mg/day) EAR-UL: 1–3 years = 2.2–7 mg; 4–8 years = 4–12 mg; 9–10 years = 7–23 mg	1–<3 years (*n* = 333)	1.6 (0.0–3.4)	35.0 (28.8–41.2)	0.1 (0.0–0.5)	35.3 (13.5–57.1)	−1.5% (0.3%)
	3–<6 years (*n* = 514)	8.9 (6.2–11.7)	21.6 (16.9–26.3)	0.5 (0.0–1.6)	20.9 (17.2–24.6)	−8.4% (−0.7%)
	6–<10 years (*n* = 479)	12.4 (8.8–15.9)	13.2 (9.7–16.8)	4.9 (2.7–7.0)	4.7 (0.0–9.5)	−7.5% (−8.5%)

EAR = Estimated Average Requirement [37]. UL = Tolerable Upper Intake Level: The highest level of daily nutrient intake that is likely to pose no risk of adverse health effects to almost all individuals in the general population. ^1^ Difference: % (usual intake < EAR) minus % (Day-1 intake < EAR). Day-1 intake: The reported 24-h recall on the first visit. Calculations were made using sample weights and the complex survey design. Usual intake: long-term daily average intake as calculated using the NCI amount-only method, using BRR weights [6].

**Table 4 nutrients-14-00285-t004:** Top 5 food items that contributed to calcium, iron and zinc intake, by age group (Day-1 intake).

Mineral	Age Group (Years)	Foods Contributing to Nutrient Intake (% Eaters, % Contribution to Total Nutrient Intake)
Calcium	Age 1–<3	Whole milk (44%, 24%), BMS (14%, 17%), Maize porridge (79%, 12%), Maas/sour milk (17%, 10%), Yoghurt (18%, 6%)
	Age 3–<6	Whole milk (49%, 24%), Maize porridge (74%, 15%), Maas/sour milk (11%, 8%), Yoghurt (14%, 7%), Pilchards/sardines (8%, 6%)
	Age 6–<10	Whole milk (48%, 22%), Pilchards/sardines (13%, 11%), Maize porridge (72%, 10%), Cheese (11%, 6%), Dairy fruit mix (11%, 5%)
Iron *	Age 1–<3	Maize porridge (79%, 30%), BMS (14%, 10%), High fiber cereals (20%, 7%), White bread (25%, 5%), Brown bread (22%, 4%)
	Age 3–<6	Maize porridge (74%, 26%), White bread (38%, 10%), Brown bread (32%, 8%), High fiber cereals (22%, 7%), Organ meat (9%, 4%)
	Age 6–<10	Maize porridge (72%, 21%), White bread (50%, 15%), Brown bread (32%, 10%), Low fiber cereals (14%, 5%), High fiber cereals (13%, 4%)
Zinc *	Age 1–<3	Maize porridge (79%, 32%), BMS (14%, 10%), Chicken (41%, 6%), Beef (11%, 6%), Whole milk (44%, 5%)
	Age 3–<6	Maize porridge (74%, 29%), Brown bread (32%, 11%), Beef (13%, 8%), Chicken (49%, 7%), White bread (38%, 7%)
	Age 6–<10	Maize porridge (72%, 24%), Brown bread (32%, 13%), White bread (50%, 11%), Beef (16%, 9%), Chicken (45%, 6%)

Eaters: children who consumed a particular food item. BMS: Breast milk substitutes. Maize porridge, white bread and brown bread are vehicles of fortification. * Nutrients used in the fortification of maize porridge, white and brown bread are iron, zinc, vitamin A, thiamine, riboflavin, niacin, folic acid and vitamin B6.

**Table 5 nutrients-14-00285-t005:** Mean (SE of the mean), median (SE of the median) and percentage differences (usual intake versus Day-1 intake) for vitamins A, C, D and E by age group.

		Age 1–<3 Years *n* = 333			Age 3–<6 Years *n* = 514			Age 6–<10 Years *n* = 479		
Nutrient		Usual Intake	Day-1 Intake	% Diff	Usual intake	Day-1 intake	% Diff	Usual Intake	Day-1 Intake	% Diff
Vitamin A (ug/day)	Mean (SE)	574.2 (67.5)	592.8 (41.5)	−3.1	607.0 (23.6)	639.2 (50.2)	−5.0	623.8 (61.7)	694.3 (58.8)	−10.2
	Median (SE)	529.5 (54.3)	367.6 (22.2)	44.0 ***	580.5 (47.8)	400.7 (19.9)	44.9 ***	550.3 (31.7)	433.2 (23.7)	27.0 ***
Vitamin C (mg/day)	Mean (SE)	47.6 (2.9)	46.6 (3.4)	2.2	39.4 (1.5)	40.8 (3.4)	−3.4	42.4 (2.9)	43.6 (3.8)	−2.8
	Median (SE)	40.2 (2.6)	32.7 (4.0)	22.9 ***	36.6 (1.4)	23.6 (2.0)	55.1 ***	37.2 (2.0)	27.3 (1.7)	36.3 ***
Vitamin D (ug/day)	Mean (SE)	2.8 (0.3)	2.9 (0.3)	−3.4	2.4 (0.1)	2.4 (0.2)	0.0	3.3 (0.1)	3.2 (0.2)	3.1
	Median (SE)	2.2 (0.3)	1.1 (0.1)	100.0 ***	2.3 (0.1)	1.2 (0.1)	91.7 ***	2.9 (0.2)	2.0 (0.2)	45.0 ***
Vitamin E (mg/day)	Mean (SE)	8.1 (0.3)	7.9 (0.5)	2.5	8.2 (0.2)	8.2 (0.4)	0.0	11.1 (0.4)	11.0 (0.5)	0.9
	Median (SE)	7.3 (0.3)	6.2 (0.3)	17.7 ***	7.5 (0.4)	6.0 (0.3)	25.0 ***	10.1 (0.5)	8.2 (0.4)	23.2 ***

Day-1 intake: The reported 24-h recall on the first visit. Usual intake: long-term daily average intake as calculated using the NCI amount-only method [6]. EAR = Estimated Average Requirement [37]. Percentage difference of the usual intake relative to day-1 intake is computed for means and medians as follows: % Diff = 100 × (Usual intake − Day-1 intake)/(Day-1 intake). There were no significant differences between the means of Day-1 intake and usual intake, *t*-test, *p*-values > 0.05. *** *p* < 0.001: Significant difference between locations of Day-1 and usual intakes, Kruskal–Wallis test.

**Table 6 nutrients-14-00285-t006:** The relationship between within-person and between-person variance for vitamins A, C, D and E as well as the CV for usual intake, by age group.

Nutrient (Box-Cox TP)	Age: 1–<3 Years				Age: 3–<6 Years				Age: 6–<10 Years			
	Var_e	Var_u	Ratio	CV(%)	Var_e	Var_u	Ratio	CV(%)	Var_e	Var_u	Ratio	CV(%)
Vitamin A (λ = 0.00)	0.48	0.15	3.20	11.8	0.67	0.07	9.57	3.9	0.45	0.24	1.88	9.9
Vitamin C (λ = 0.29)	5.93	4.31	1.38	6.0	7.3	0.98	7.45	3.8	5.59	2.68	2.09	6.7
Vitamin D (λ = 0.26)	1.82	1.11	1.64	11.5	2.71	0.01	271.00	5.1	2.14	0.67	3.19	4.2
Vitamin E (λ = 0.14)	0.93	0.32	2.91	3.5	0.78	0.36	2.17	2.9	0.77	0.44	1.75	3.4

A lambda value < 0.15 reflects a larger mean bias as a result of sensitivity to the transformation applied [16]. Lambda values associated with the first execution or base run of the macros are reported. Var_e: Within-person variance; Var_u: Between person variance; Ratio = Var_e/Var_u, the ratio of within-person to between-person variance. CV: Coefficient of variation of the mean of usual intake calculated as follows: CV = 100 × (standard error of the mean)/mean. TP: Transformation parameter.

**Table 7 nutrients-14-00285-t007:** Comparison of percentage below EAR and above UL cut-points for risk of vitamins A, D, E and C deficiency and excess respectively, by age group.

		Day-1 Intake		Usual Intake		Difference ^1^ for % < EAR & (% < UL)
		% < EAR	% > UL	% < EAR	% > UL	
Vitamin A (ug/day) EAR-UL: 1–3 years = 210–600 ug; 4–8 years = 275–900 ug; Male: 9–<10 years = 445–1700 ug; Female: 9–<10 years = 420–1700 ug	1–<3 years (*n* = 333)	16.1 (11.5–20.7)	27.8 (22.3–33.3)	1.2 (0.0–4.1)	37.5 (10.3–64.7)	14.9% (9.7%)
	3–<6 years (*n* = 514)	24.5 (19.7–29.4)	18.7 (13.6–23.9)	0.4 (0.0–4.2)	21.5 (16.1–27.0)	24.1% (2.8%)
	6–<10 years (*n* = 479)	29.3 (24.9–33.8)	12.1 (8.4–15.8)	12.0 (6.9–17.0)	13.7 (0.0–27.4)	17.3% (1.6%)
Vitamin C (mg/day) EAR-UL: 1–3 years = 13–400 mg; 4–8 years = 22–650 mg; 9–<10 years = 39–1200 mg	1–<3 years (*n* = 333)	21.3 (14.8–27.8)	0.0 (-)	7.4 (3.0–11.8)	0.0 (-)	13.9% (0.0%)
	3–<6 years (*n* = 514)	39.1 (33.0–45.2)	0.0 (-)	9.0 (0.0–20.0)	0.0 (-)	30.1% (0.0%)
	6–<10 years (*n* = 479)	40.3 (34.4–46.2)	0.0 (-)	25.3 (11.8–38.8)	0.0 (-)	15.0% (0.0%)
Vitamin D (ug/day) EAR-UL: 1–3 years = 10–63 ug; 4–8 years = 10–75 ug; 9–<10 years = 10–100 ug	1–<3 years (*n* = 333)	94.3 (90.8–97.8)	0.0 (-)	98.2 (96.9–99.4)	0.0 (-)	3.9% (0.0%)
	3–<6 years (*n* = 514)	96.5 (94.3–98.7)	0.0 (-)	100.0 (-)	0.0 (-)	3.5% (0.0%)
	6–<10 years (*n* = 479)	93.8 (91.0–96.5)	0.0 (-)	99.3 (98.3–100.0)	0.0 (-)	5.5% (0.0%)
Vitamin E (mg/day) EAR-UL: 1–3 years = 5–90 mg; 4–8 years = 6–135 mg; 9–<10 years = 9–270 mg	1–<3 years (*n* = 333)	36.2 (29.5–43.0)	0.0 (-)	18.2 (7.9–28.4)	0.0 (-)	−18.0% (0.0%)
	3–<6 years (*n* = 514)	46.6 (41.7–51.5)	0.0 (-)	26.9 (10.2–43.5)	0.0 (-)	−19.7% (0.0%)
	6–<10 years (*n* = 479)	35.2 (29.9–40.5)	0.0 (-)	18.8 (8.5–29.2)	0.0 (-)	16.4% (0.0%)

EAR = Estimated Average Requirement [37]. UL = Tolerable Upper Intake Level: The highest level of daily nutrient intake that is likely to pose no risk of adverse health effects to almost all individuals in the general population. ^1^ Difference: % (usual intake <EAR) minus % (Day-1 intake < EAR). Day-1 intake: The reported 24-h recall on the first visit. Calculations were made using sample weights and the complex survey design. Usual intake: long-term daily average intake as calculated using the NCI amount-only method using BRR weights [6].

**Table 8 nutrients-14-00285-t008:** Top 5 food items that contributed to vitamins A, C, D and E intakes, by age group (Day-1 intake).

Vitamin	Age Group (Years)	Foods Contributing to Nutrient Intake (% Eaters, % Contribution to Total Nutrient Intake)
Vitamin A *	Age 1–<3	Maize porridge (79%, 27%), Vegetables-carotene (other) (9%, 14%), Organ meat (5%, 11%), BMS (14%, 10%), Whole milk (44%, 7%)
	Age 3–<6	Organ meat (9%, 32%), Maize porridge (74%, 22%), Vegetables-carotene (other) (10%, 12%), Whole milk (49%, 5%), PUM fat (28%, 4%)
	Age6–<10	Organ meat (9%, 27%), Maize porridge (72%, 22%), Vegetables-carotene (other) (9%, 10%), White bread (50%, 7%), PUM fat (35%, 7%)
Vitamin C	Age 1–<3	Fruit fresh vitamin C rich (12%, 15%), BMS (14%, 15%), Potato/sweet potato (33%, 14%), Fruit juice (6%, 14%), Vegetables- vitamin C rich (24%, 8%)
	Age 3–<6	Fruit juice (7%, 18%), Potato/sweet potato (31%, 16%), Fruit fresh vitamin C (8%, 16%), Vegetables-vitamin C rich (28%, 12%), Maize porridge (74%, 9%)
	Age 6–<10	Fruit juice (7%, 23%), Fruit fresh-vitamin C rich (9%, 17%), Potato/sweet potato (33%, 15%), Vegetables-vitamin C rich (31%, 13%), Maize porridge (72%, 5%)
Vitamin D	Age 1–<3	BMS (14%, 30%), eggs (14%, 25%), Pilchards/sardines (6%, 16%), PUM fat (21%, 5%), Whole milk (44%, 3%)
	Age 3–<6	Eggs (11%, 26%), Pilchards/sardines (8%, 25%), PUM fat (28%, 12%), Dairy fruit mix (13%, 5%), Chicken (49%, 4%)
	Age 6–<10	Pilchards/sardines (13%, 32%), Eggs (12%, 23%), PUM fat (35%, 13%), Fat cakes (7%, 4%), Cereal low fiber (14%, 4%)
Vitamin E	Age 1–<3	PU fat/oil (12%, 15%), Maize porridge (79%, 11%), BMS (14%, 11%), Salty snacks (44%, 8%), PUM fat (21%, 7%)
	Age 3–<6	PU fat/oil (14%, 17%), PUM fat (28%, 13%), Maize porridge (74%, 11%), Salty snacks (48%, 9%), Vegetables- vitamin C (28%, 6%)
	Age 6–<10	PU fat/oil (18%, 18%), PUM fat (35%, 17%), Salty snacks (54%, 9%), Maize porridge (72%, 7%), Fat cakes (7%, 7%)

Eaters: children who consumed a particular food item. Vegetables-carotene (other): carrots, mixed vegetables containing mostly carrots; BMS: Breast milk substitutes; Vegetables—carotene (green leaves): spinach varieties; PUM fat: polyunsaturated margarine /medium fat spread; Fruit fresh-vitamin C rich: citrus fruit, guava, green melon, strawberry, pineapple; Vegetable—vitamin C: Cabbage, broccoli, cauliflower, peppers, tomato; PU fat/oil: Vegetable oil, sunflower oil; fat cakes are deep fried balls of bread dough; Salty snacks: Maize-based snacks, popcorn, potato crisps. Maize porridge, white bread and brown bread are vehicles of fortification. * Nutrients used in fortification of maize porridge, white and brown bread are iron, zinc, vitamin A, thiamine, riboflavin, niacin, folic acid and vitamin B6.

**Table 9 nutrients-14-00285-t009:** Mean (SE of the mean), median (SE of median) and percentage differences (usual intake versus Day-1 intake) for thiamine, niacin, riboflavin, vitamin B6, vitamin B12 and folate intakes, by age group.

		Age 1–<3 Years *n* = 333			Age 3–<6 Years *n* = 514			Age 6–<10 Years *n* = 479		
Nutrient		Usual Intake	Day-1 Intake	% Diff	Usual Intake	Day-1 Intake	% Diff	Usual Intake	Day-1 Intake	% Diff
Thiamine (mg/day)	Mean (SE)	1.0 (0.03)	1.0 (0.04)	0.0	1.0 (0.01)	1.0 (0.03)	0.0	1.2 (0.03)	1.2 (0.02)	0.0
	Median (SE)	0.9 (0.01)	0.9 (0.04)	0.0	1.0 (0.02)	0.9 (0.03)	11.1 ***	1.1 (0.03)	1.1 (0.04)	0.0 *
Niacin (mgNE/day)	Mean (SE)	11.5 (0.4)	11.5 (0.4)	0.0	14.2 (0.2)	14.2 (0.4)	0.0	17.2 (0.4)	17.3 (0.4)	−0.6
	Median (SE)	11.3 (0.3)	10.6 (0.4)	6.6 **	13.8 (0.2)	13.2 (0.4)	4.6 **	16.8 (0.4)	16.7 (0.5)	0.6 *
Riboflavin (mg/day)	Mean (SE)	0.9 (0.1)	0.9 (0.04)	0.0	0.9 (0.02)	0.9 (0.03)	0.0	1.0 (0.03)	1.0 (0.04)	0.0
	Median (SE)	0.8 (0.1)	0.8 (0.04)	0.0 **	0.9 (0.02)	0.8 (0.05)	12.5 ***	0.9 (0.02)	0.9 (0.04)	0.0 ***
Vitamin B_6_ (mg/day)	Mean (SE)	1.4 (0.05)	1.4 (0.05)	0.0	1.8 (0.01)	1.8 (0.04)	0.0	2.5 (0.04)	2.5 (0.1)	0.0
	Median (SE)	1.3 (0.04)	1.2 (0.1)	8.3 *	1.8 (0.02)	1.7 (0.05)	5.9 ***	2.4 (0.04)	2.2 (0.1)	9.1 **
Vitamin B_12_ (ug/day)	Mean (SE)	2.2 (0.1)	2.3 (0.3)	−4.4	2.9 (0.1)	3.3 (0.4)	−12.1	4.3 (0.4)	4.7 (0.6)	−8.5
	Median (SE)	1.7 (0.5)	1.1 (0.1)	54.6 ***	2.9 (0.2)	1.3 (0.1)	123.1 ***	3.5 (0.3)	1.7 (0.1)	105.8 ***
Folate (ug/day)	Mean (SE)	225.4 (12.7)	225.0 (12.1)	0.2	253.2 (4.0)	253.2 (11.6)	0.0	282.1 (7.5)	284.6 (7.9)	−0.9
	Median (SE)	210.1 (10.5)	200.0 (9.8)	5.1 **	238.7 (9.9)	202.3 (11.8)	18.0 ***	266.0 (5.5)	242.9 (5.5)	9.5 **

Day-1 intake: The reported 24-h recall on the first visit. Usual intake: long-term daily average intake as calculated using the NCI amount-only method [6]. EAR = Estimated Average Requirement [37]. Percentage difference of the usual intake relative to Day-1 intake is computed for means and medians as follows: % Diff = 100 × (Usual intake − Day-1 intake)/(Day-1 intake). There were no significant differences between the means of Day-1 intake and usual intake, *t*-test, *p*-values > 0.05. * *p* < 0.05; ** *p* < 0.01; *** *p* < 0.001: Significant difference between locations of Day-1 and usual intakes, Kruskal–Wallis test.

**Table 10 nutrients-14-00285-t010:** Within-person and between-person variance, ratio of these two parameters and the coefficient of variation for usual intake of thiamine, niacin, riboflavin, vitamin B6, vitamin B12 and folate, by age group.

Nutrient (Box-Cox TP)	Age: 1–<3 Years				Age: 3–<6 Years				Age: 6–<10 Years			
	Var_e	Var_u	Ratio	CV (%)	Var_e	Var_u	Ratio	CV (%)	Var_e	Var_u	Ratio	CV (%)
Thiamine (λ = 0.26)	0.11	0.11	1.00	2.9	0.12	0.07	1.71	1.3	0.10	0.08	1.25	2.9
Niacin (λ = 0.34)	1.06	0.19	5.58	3.2	0.76	0.44	1.73	1.7	0.85	0.54	1.57	2.2
Riboflavin (λ = 0.18)	0.22	0.16	1.38	9.8	0.24	0.06	4.00	2.7	0.17	0.15	1.13	2.7
Vitamin B6 (λ = 0.20)	0.21	0.06	3.50	3.4	0.25	0.06	4.17	0.8	0.23	0.13	1.77	1.5
Vitamin B12 (λ = 0.13)	1.27	0.62	2.05	5.9	2.48	0.03	2.67	4.2	2.01	0.71	2.83	9.0
Folate (λ = 0.07)	0.44	0.28	1.57	5.7	0.52	0.23	2.26	1.6	0.38	0.23	1.65	2.7

A lambda value < 0.15 reflects a larger mean bias as a result of sensitivity to the transformation applied [16]. Lambda values associated with the first execution or base run of the macros are reported. Var_e: Within-person variance; Var_u: Between person variance; Ratio = Var_e/Var_u, the ratio of within-person to between-person variance. CV: Coefficient of variation of the mean of usual intake calculated as follows: CV = 100 × (standard error of the mean)/mean. TP: Transformation parameter.

**Table 11 nutrients-14-00285-t011:** Comparison of percentage below EAR and above UL cut-points for risk of thiamine, niacin, riboflavin, vitamin B6, vitamin B12 and folate deficiency and excess (where applicable), by age group.

		Day-1 Intake		Usual Intake		Difference ^1^ in % < EAR & % > UL
		% < EAR	% > UL	% < EAR	% > UL	
Thiamine (mg/day)EAR: 1–3 years = 0.4 mg; 4–8 years = 0.5 mg; 9–<10 years = 0.7 mg; No UL	1–<3 years (*n* = 333)	4.8 (1.5–8.0)	-	1.6 (0.0–4.8)	-	3.2%
	3–<6 years (*n* = 514)	6.9 (4.0–9.7)		0.7 (0.0–2.7)		6.2%
	6–<10 years (*n* = 479)	4.8 (2.0–7.6)		1.3 (0.0–3.1)		3.5%
Niacin (mgNE/day) EAR-UL: 1–3 years = 5–10 mgNE; 4–8 years = 6–15 mgNE; 9–<10 years = 9–20 mgNE	1–<3 years (*n* = 333)	10.7 (6.6–14.9)	56.0 (49.5–62.4)	0.1 (0.0–0.9)	0.0 (-)	10.6% (−56.0%)
	3–<6 years (*n* = 514)	5.9 (3.3–8.5)	51.2 (46.5–55.9)	0.3 (0.0–1.0)	0.0 (-)	5.6% (−51.2%)
	6–<10 years (*n* = 479)	5.4 (3.1–7.7)	56.8 (51.7–61.8)	0.5 (0.2–0.8)	0.0 (-)	4.9% (−56.8%)
Riboflavin (mg/day) EAR: 1–3 years = 0.4 mg; 4–8 years = 0.5 mg; 9–<10 years = 0.8 mg; No UL	1–<3 years (*n* = 333)	17.4 (11.2–23.7)		5.6 (0.0–12.6)		11.8%
	3–<6 years (*n* = 514)	19.5 (14.3–24.7)		2.3 (0.0–8.1)		17.2%
	6–<10 years (*n* = 479)	23.4 (18.5–28.2)		11.4 (4.9–18.0)		12.0%
Vitamin B_6_ (mg/day) EAR-UL: 1–3 years = 0.4–30 mg; 4–8 years = 0.5–40 mg; 9–<10 years = 0.8–60 mg	1–<3 years (*n* = 333)	2.5 (0.7–4.3)	0.0 (-)	0.0 (-)	0.0 (-)	2.5% (0.0%)
	3–<6 years (*n* = 514)	2.6 (0.9–4.3)	0.0 (-)	0.0 (-)	0.0 (-)	2.6% (0.0%)
	6–<10 years (*n* = 479)	0.8 (0.0–1.6)	0.0 (-)	0.0 (-)	0.0 (-)	0.8% (0.0%)
Vitamin B_12_ (ug/day) EAR: 1–3 years = 0.7 ug; 4–8 years = 1.0 ug; 9–<10 years = 1.5 ug; No UL	1–<3 years (*n* = 333)	34.4 (27.0–41.8)		14.3 (0.0–56.4)		20.1%
	3–<6 years (*n* = 514)	36.7 (30.9–42.4)		0.0 (-)		36.7%
	6–<10 years (*n* = 479)	35.0 (29.6–40.3)		5.4 (0.0–13.2)		29.6%
Folate (ug/day) EAR-UL: 1–3 years = 120–300 ug; 4–8 years = 160–400 ug; 9–<10 years = 250–600 ug	1–<3 years (*n* = 333)	22.9 (16.6–29.3)	20.8 (15.5–26.0)	9.6 (0.0–22.1)	18.6 (6.1–31.2)	13.3% (−2.4%)
	3–<6 years (*n* = 514)	26.5 (21.1–31.9)	21.6 (16.1–27.1)	10.3 (0.0–22.2)	14.6 (9.2–19.9)	16.2% (−7.0%)
	6–<10 years (*n* = 479)	27.8 (23.4–32.2)	16.2 (11.5–20.9)	15.5 (12.4–18.6)	11.1 (6.4–15.9)	12.3% (−4.4%)

EAR = Estimated Average Requirement [37]. UL = Tolerable Upper Intake Level: The highest level of daily nutrient intake that is likely to pose no risk of adverse health effects to almost all individuals in the general population. ^1^ Difference: % (usual intake < EAR) minus % (Day-1 intake < EAR). Day-1 intake: The reported 24-h recall on the first visit. Calculations were made using sample weights and the complex survey design. Usual intake: long-term daily average intake as calculated using the NCI amount-only method, using BRR weights.

**Table 12 nutrients-14-00285-t012:** Top 5 food items that contributed to thiamine, niacin, riboflavin, vitamin B6, vitamin B12 and folate intakes, by age group (Day-1 intake).

Vitamin	Age Group (Years)	Foods Contributing to Nutrient Intake (% Eaters, % Contribution to Total Nutrient Intake)
Thiamine *	Age 1–<3	Maize porridge (79%, 42%), BMS (14%, 9%), High fiber cereal (20%, 6%), Potato/sweet potato (33%, 4%), Brown bread (22%, 4%)
	Age 3–<6	Maize porridge (74%, 39%), Brown bread (32%, 8%), High fiber cereal (22%, 7%), White bread (38%, 7%), Potato/sweet potato (31%, 4%)
	Age 6–<10	Maize porridge (72%, 35%), White bread (50%, 11%), Brown bread (32%, 10%), Low fiber cereal (14%, 5%), Processed meat (32%, 4%)
Niacin *	Age 1–<3	Maize porridge (79%, 26%), Chicken (41%, 20%), High fiber cereal (20%, 7%), Brown bread (22%, 6%), White bread (25%, 5%)
	Age 3–<6	Chicken (49%, 20%), Maize porridge (74%, 20%), Brown bread (32%, 10%), White bread (38%, 10%), High fiber cereal (22%, 7%)
	Age 6–<10	Maize porridge (72%, 17%), Chicken (45%, 17%), White bread (50%, 15%), Brown bread (32%, 12%), Pilchards/sardines (13%, 6%)
Riboflavin *	Age 1–<3	Maize porridge (79%, 17%), BMS (14%, 14%), Whole milk (44%, 14%), High fiber cereal (20%, 9%), Maas/sour milk (17%, 4%)
	Age 3–<6	Maize porridge (74%, 17%), Whole milk (49%, 12%), High fiber cereal (22%, 10%), Organ meat (9%, 10%), Chicken (49%, 5%)
	Age 6–<10	Maize porridge (72%, 15%), Whole milk (48%, 10%), Organ meat (9%, 8%), Low fiber cereal (14%, 8%), High fiber cereal (13%, 6%)
Vitamin B6 *	Age 1–<2	Maize porridge (79%, 33%), White bread (25%, 12%), Brown bread (22%, 12%), Potato/sweet potato (33%, 6%), BMS (14%, 5%)
	Age 3–<6	Maize porridge (74%, 24%), White bread (38%, 22%), Brown bread (32%, 20%), Potato/sweet potato (31%, 6%), Chicken (49%, 3%)
	Age 6–<10	White bread (50%, 31%), Brown bread (32%, 21%), Maize porridge (72%, 18%), Potato/sweet potato (33%, 5%), Low fiber cereal (14%, 3%)
Vitamin B_12_	Age 1–<2	Pilchards/sardines (6%, 32%), Whole milk (44%, 15%), Organ meat (5%, 11%), Eggs (14%, 6%), Beef (11%, 6%)
	Age 3–<6	Organ meat (9%, 42%), Pilchards/sardines (8%, 24%), Whole milk (49%, 8%), Beef (13%, 6%), Eggs (11%, 3%)
	Age 6–<10	Pilchards/sardines (13%, 36%), Organ meat (9%, 30%), Beef (16%, 8%), Whole milk (48%, 6%), Eggs (12%, 3%)
Folate *	Age 1–<2	Maize porridge (79%, 56%), BMS (14%, 5%), Brown bread (22%, 5%), Organ meat (5%, 5%), White bread (25%, 4%)
	Age 3–<6	Maize porridge (74%, 48%), Organ meat (9%, 9%), Brown bread (32%, 9%), White bread (38%, 7%), Low fiber cereal (10%, 3%)
	Age 6–<10	Maize porridge (72%, 43%), White bread (50%, 12%), Brown bread (32%, 11%), Organ meat (9%, 7%), Low fiber cereal (14%, 4%)

BMS: Breast milk substitutes; Salty snacks: Maize-based snacks, popcorn, potato crisps. Maize porridge, white bread and brown bread are vehicles of fortification. * Nutrients used in fortification of maize porridge, white and brown bread are iron, zinc, vitamin A, thiamine, riboflavin, niacin, folic acid, and vitamin B6.

## Data Availability

The data presented in this study are available on request from the corresponding author pending ethical approval from the Faculty of Health Sciences Human Research Ethics Committee, University of Cape Town.

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
