# Peer review of "Illustration of the Importance of Adjustment for within- and between-Person Variability in Dietary Intake Surveys for Assessment of Population Risk of Micronutrient Deficiency/Excess Using an Example Data Set"

_nutrients, 2022, doi:10.3390/nu14020285_

Round 1

Reviewer 1 Report

This article is well structured and covers a very interesting topic. Each section is well-argued: The introduction lays out the theoretical context and designated goal well. The methodology is well explained and the models used lend themselves well. The results, including through the use of tables, are clear and comprehensive, as are the discussions. The only thing I would propose to improve is the conclusions. You could emphasize more the theoretical, policy, and managerial implications your results provided.

Reviewer 2 Report

The paper by Nel  et al. describes a re-analysis of a previously published dataset that has already been covered in three publications. The first paper is the most closely related, focusing on micronutrients, the second focusing on macronutrients and the third paper on sociodemographic factors, all using the same survey.

1: Senekal M, Nel J, Malczyk S, Drummond L, Steyn NP. Provincial Dietary Intake

Study (PDIS): Micronutrient Intakes of Children in a Representative/Random

Sample of 1- to <10-Year-Old Children in Two Economically Active and Urbanized

Provinces in South Africa. Int J Environ Res Public Health. 2020 Aug

14;17(16):5924. doi: 10.3390/ijerph17165924. PMID: 32824083; PMCID: PMC7460187.

2: Steyn NP, Nel JH, Malczyk S, Drummond L, Senekal M. Provincial Dietary Intake

Study (PDIS): Energy and Macronutrient Intakes of Children in a

Representative/Random Sample of 1-<10-Year-Old Children in Two Economically

Active and Urbanized Provinces in South Africa. Int J Environ Res Public Health.

2020 Mar 5;17(5):1717. doi: 10.3390/ijerph17051717. PMID: 32151074; PMCID:

PMC7084522.

3: Senekal M, Nel JH, Malczyk S, Drummond L, Harbron J, Steyn NP. Provincial

Dietary Intake Study (PDIS): Prevalence and Sociodemographic Determinants of the

Double Burden of Malnutrition in A Representative Sample of 1 to Under 10-Year-

Old Children from Two Urbanized and Economically Active Provinces in South

Africa. Int J Environ Res Public Health. 2019 Sep 10;16(18):3334. doi:

10.3390/ijerph16183334. PMID: 31509998; PMCID: PMC6765782.

Already in the paper #1 above the authors cite as a strength of their paper “A further strength is that we completed two additional 24-h recalls in a 9% subsample to apply the National Cancer Institute (NCI) method [17,18] to obtain results that are more representative of usual intake.”

The current paper describes an in-depth analysis of the differences between micronutrient uptake based on single 24 hour recall versus “usual” recall, which takes into account data (that was already collected for the previous three publications) to correct for variability due to the single datapoint by extrapolating from the roughly 10% sub-sample for which 2nd and 3rd recall data are available.

The tables demonstrate that the differences are relatively small in most cases and the graphical representations also show that the distributions do not substantially distract from the major conclusions in terms of distance to the recommended intake values.

The major conclusions from looking at this data is the deficiency in non-fortified micronutrients Vitamin D and calcium and these are exactly the two nutrients pointed out also in the earlier study. Inter- and intraperson variability  doesn’t change the fact that they are under-represented in the diet.

Because there is relatively little new scientific insights gained from this study, my recommendation is that as this is a highly methods focused paper, it should be rewritten as such. The methods section is very detailed and therefore suitable for replication by others with similarly specialized interest. Instead of declaring this study as an “example” in the title, I think the title should contain reference to the methodological advanced made by in depth application of the NCI method.

Round 2

Reviewer 2 Report

the authors have addressed my concerns